# Beyond words: Relationships between emoji use, attachment style, and emotional intelligence

**Simon Dubé**[1]*, **Amanda N. Gesselman**[1], **Ellen M. Kaufman**[1,2], **Margaret Bennett-Brown**[1,3], **Vivian P. Ta-Johnson**[4¤], **Justin R. Garcia**[1,5]

1 The Kinsey Institute, Indiana University, Bloomington, Indiana, 2 Luddy School of Informatics, Computing and Engineering, Indiana University, Bloomington, Indiana, 3 Department of Communication Studies, Texas Tech University, Lubbock, Texas, United States of America, 4 Department of Psychology, Lake Forest College, Lake Forest, Illinois, United States of America, 5 Department of Gender Studies, Indiana University, Bloomington, Indiana

¤ Current address: Department of Psychology, University of Texas at San Antonio, San Antonio, Texas, United States of America

* simondube.ta@gmail.com

## Abstract

Assessing the relationships between emoji use and traits related to communication and interpersonal skills can provide insights into who employs emojis and the psychological mechanisms underlying computer-mediated communications. This online study investigated associations between emoji use frequency, attachment style, and emotional intelligence across genders and relationship types in a Mechanical Turk sample of 320 adults (≥18y; 191 women, 123 men, and 4 transgender individuals). Correlational analyses showed that emotional intelligence was positively related to emoji use with friends, while avoidant attachment was negatively related to emoji use with friends and dating or romantic partners. This pattern of associations varied across genders and relationship types, with women using emojis more frequently than men with friends and family. Such findings suggest that individuals higher on emotional intelligence with secure attachment may employ emojis more frequently across contexts where more conventional non-verbal cues are lacking. These findings are important given the prevalence of virtual communications in our everyday lives.

## 1. Introduction

Emojis are characters depicting emotions, objects, animals, and more [1]. They can be sent via computers or smartphones—alone or with text—to create more complex meaning during virtual communications [2]. Emoji options have had multiple iterations and updates, appear to be used widely, and may transcend translation barriers of written language. Emojis have become a source of interest in computer-mediated communication (CMC) as they can support a text-based message and meet a need for non-verbal support elements in the absence of signals and cues from face-to-face communication.

Previous research on emojis and their precursors—emoticons—stems from various disciplines, such as computer science, marketing, communication, behavioral science, linguistics,

**Data Availability Statement:** Data available at the Open Science Framework (OSF): https://osf.io/2gzu7/.

**Funding:** The author(s) received no specific funding for this work.

**Competing interests:** The authors have declared that no competing interests exist.

psychology, medicine, and education (see [1,2] for reviews). This research mostly focuses on why and how people use these characters (e.g., motivations and frequency), their impact on communication across contexts (e.g., message perception and facilitating/impairing understanding), their relationships to individual characteristics, such as gender or personality traits, as well as how they may reveal or help understand human emotions [1,2].

For example, this research suggests that people often send emojis to reduce uncertainty, modulate the valence and tone of a message, or increase the precision of a message [3–5], and while they tend to do so, these characters can also be misunderstood depending on contexts, features, and cultural background [6]. This research also suggests that men tend to use a wider range of emoji types, but also more consistently punctuate emotional expressions with the same characters [7,8]. Women, on the other hand, tend to use emojis more frequently and positively, including in public versus private communications [5,7,9], as well as perceive their use as clearer and more familiar [10]. Furthermore, personality traits such as extraversion and neuroticism (i.e., low emotional stability) respectively negatively and positively relate to using emojis in order to avoid awkwardness, while agreeableness positively relates to using emojis in order to express emotions, clarify or disambiguate messages, lighten the mood, and convey a sense of humor [11].

In parallel, a significant body of research in the transdisciplinary field of affective computing has employed (increasingly refined) data analysis, artificial intelligence, and machine learning techniques to recognize and interpret affective states through emojis [1,2]. For instance, using unsupervised learning, some researchers automatically trained sentiment analysis models on emoji datasets [12], while others developed models to classify and analyze Twitter message sentiments [13]. More recent work also explores the accuracy of machine learning and deep learning methods in predicting relevant emoji based on emotional and contextual text cues [14].

There is also some evidence that the interpersonal context of emoji use matters. In hypothetical scenarios, participants prefer to receive them from closer social connections versus more distant or business associates [15]. Emoji use has been associated with more romantic and sexual interactions, and with maintaining connection after a first date [16]. In studies of email correspondence from institutional leaders, emojis are associated with positive perceptions of likability and effectiveness for men, but viewed as positive but less appropriate for women [17]. These studies demonstrate both contextual and gender-based differences in frequency and type of use of emojis, along with potential biases towards emoji use (e.g., based on the user's gender).

Despite the pervasiveness of CMC and emoji use in our daily social lives, relatively little is known about *who* uses emojis beyond evidence of differences related to gender and personality traits. However, individual trait-like characteristics pertaining to communication skills and interpersonal relationships, such as attachment styles and emotional intelligence, may also affect how emojis are used. A better understanding of these associations can provide insights into the profile of emoji users, and the psychological mechanisms underlying the way emojis are leveraged as a communication tool.

## 1.1. Attachment theory

Developed by Bowlby [18], attachment theory originally examined patterns of parent–child bonding. Bowlby's observational and experimental research, along with contributions from other scholars, revealed distinct styles of bonding in children that predictably led to different behavioral outcomes [18]. These styles were broadly classified into three types, as documented cross-culturally by Ainsworth and colleagues [19]: anxious, avoidant, and secure attachment.

Anxious attachment, characterized by pronounced distress upon reunion with one's caregiver following a brief separation (e.g., tantrums), is believed to arise from inconsistent caregiver responsiveness [19]. Conversely, avoidant attachment—characterized by indifference or subdued affect during reunion with one's caregiver following a brief separation—may stem from emotionally detached or unresponsive caregiving, and reflect the child's self-reliance and distrust in the caregiver's support [19]. Both anxious and avoidant attachment styles indicate a child's lack of felt security within the relationship with their primary caregiver. In contrast, children with secure attachment styles were enthusiastic in their reunions with their caregivers, following a brief separation [19]. As a result of more responsive and nurturing caregiving, securely attached children can more effectively rely on their caregivers, and thus develop an understanding of close others as a source of reliable support [19].

Building on the foundations of attachment theory, researchers extended its application to understand and explain how people behave in relationships across a range of interpersonal contexts, including but not limited to family members, co-workers, and romantic partners [20–24]. The guiding principle being that early parent–child bonding experiences form basic relational patterns that manifest themselves, later in life, in other relationships. With regard to romantic relationships, for instance, researchers observed that the bond between adult partners shares key characteristics with the parent–child bond, likely stemming from a similar motivational system (cf., [25]). These characteristics include a sense of security when a partner is near and responsive, distress when one must be physically separate from one's partner, and intimate behaviors, such as "baby talk," eye-gazing, and cuddling [26]. Like children, adults also tend to share discoveries and new experiences with their partners [21,25,27].

Further investigations categorized adults into similar attachment styles as children [21,28,29]. Adult attachment styles are typically assessed along two dimensions: anxiety (i.e., fear of abandonment and uncertainty about a partner's availability) and avoidance (i.e., discomfort with emotional closeness). Those with high levels of anxious attachment—sometimes referred to as 'anxiously attached'—often experience fears of abandonment, leading to maladaptive behaviors aimed at maintaining closeness with their partner. Conversely, adults higher in avoidant attachment—sometimes referred to as 'avoidantly attached'—tend to maintain emotional distance from their romantic partners and minimize opportunities for emotional intimacy. An adult with low levels of both anxious and avoidant attachment are categorized as securely attached. These individuals display confidence in their partner's responsiveness and are comfortable with emotional intimacy.

Adult attachment styles significantly influence a spectrum of outcomes in romantic relationships. A comprehensive meta-analysis encompassing more than 20,000 participants revealed that both anxious and avoidant attachment styles negatively impact romantic relationship quality [30]. Higher levels of anxious attachment were specifically associated with more frequent conflict between partners. However, avoidant attachment was associated with a broader range of negative outcomes, including less social support, feelings of social disconnection, and lower relationship satisfaction overall. Other studies have shown that avoidant attachment is associated with lower levels of trust, less investment in one's relationship, and aversion to commitment—likely due to more avoidantly attached individuals' perceptions of emotional closeness as personally risky [31–34].

While the bulk of research on adult attachment styles has primarily focused on romantic relationships, their influence extends to various social interactions, such as relationships with family members, co-workers, and friends [20–24]. For example, studies suggest that the quality of parent-child relationships positively correlates with the quality of sibling relationships during emerging adulthood [23] and that employees with a secure attachment style tend to develop more productive relationships with their co-workers than those with insecure

attachment [24,35,36]. Studies further suggest that secure attachment increases the likelihood of developing positive peer relationships characterized by intimacy, trust, and effective communication [20]. In contrast, insecure attachment styles are often associated with lower social skills, which may in turn impair the formation of close and supportive friendships [20].

There is also evidence that higher levels of anxious and avoidant attachment correlate with lower relational satisfaction in both romantic and professional domains [37,38]. These attachment styles also impact the level of companionship and intimacy in friendships, with more anxious or avoidant attachment often leading to increased conflict [39,40]. Furthermore, individuals with more anxious or avoidant attachment styles experience challenges in clearly expressing their needs within social relationships [41,42]. This contributes to negative affect for the individual and their social partner(s), and is associated with less effective conflict resolution [43,44]. These individuals also tend to engage in less self-disclosure and supportive communication [44–46], affecting interactions across various contexts including friendly, romantic, professional, and even medical relationships (cf., [42,47,48]).

A growing literature has demonstrated the influence of attachment styles in the context of CMC as well. Anxious attachment often leads to negative interpretations of text-based communication, a medium prone to ambiguity and misinterpretation due to the absence of nonverbal cues [49,50]. Individuals with higher levels of avoidant attachment exhibit lower communication competence and a preference for more distanced forms of communication (e.g., text-based communication), rather than communication that allows for greater closeness (e.g., face-to-face communication; [49,51,52]). While these studies suggest that attachment styles may affect how people interact with others both in-person and via text-based messaging, the role of attachment in emoji use has remained unexplored. Because emojis may be employed as a tool or strategy for enhancing the expression of emotions and enriching the emotional context of a message, attachment styles are likely to evoke differences in the frequency or particular use of emojis. In the current study, we examined differences in emoji use with varying social partners as a function of participants' attachment styles. To further understand differences in emoji use as a function of individual traits that may impact the expression and/or interpretation of emotionally-laden messages, we also examined the moderating impact of emotional intelligence.

## 1.2. Emotional intelligence

Emotional intelligence (EI) refers to the ability to process and manage one's emotions and those of others [53]. EI involves a combination of self and social awareness, as well as self and relationship management, to navigate interpersonal situations [54–56]. As such, people with higher EI have been shown to communicate, handle conflicts, and build rapport with others more effectively than people with lower EI [57–59]. In face-to-face communication, high EI individuals can detect emotion from facial cues more accurately than those with low EI [60]. As a result, people with higher EI have more positive interpersonal interactions with partners and create more satisfying relational environments (cf., [61]).

In virtual communications, social cues are fewer. As such, possessing higher EI in CMC contexts may be advantageous for understanding and managing CMC-based social interactions. Research conducted from education and business managerial standpoints have supported this idea. For example, employees in virtual team environments with higher EI were found to have better quality communication and greater team success [62]. Similarly, students in an online classroom environment with higher EI showed a greater willingness to communicate compared to lower EI peers [63]. These studies, along with others in similar disciplines (e.g., [64]), suggest that EI may drive differences in online communication patterns. In a study

examining EI and emoji use, it was found that participants with higher EI interpreted emoji-laden messages as self-revelatory, while those with lower EI viewed them as conveying factual information [65]. This indicates that individuals with higher EI are better at interpreting the emotional cues in emojis. These findings on EI's role in virtual communication pave the way for a deeper exploration into how EI intersects with other psychological frameworks, particularly attachment. Understanding this intersection could be crucial in comprehending the nuances of how people form and maintain relationships in an increasingly digital world.

A number of studies have documented associations between attachment styles and EI. These studies largely report that lower levels of anxious or avoidant attachment—and thus more secure attachment style—are associated with higher levels of EI [66]. Individuals with more secure attachment styles are generally better at understanding and managing their own emotions and those of others, likely due to their perceptions of social partners as sources of comfort and support. However, those with more anxious attachment styles may struggle with emotion regulation and heightened emotional sensitivity [67], impairing their ability to accurately perceive and respond to emotions. Similarly, individuals with more avoidant attachments are typically uncomfortable with emotional displays, and may find it challenging to recognize and/or empathize with emotional states. Taken together, research suggests that adults with higher levels of anxious or avoidant attachment may also have limited emotional intelligence.

However, while attachment styles are viewed as generally stable over one's lifetime (cf. [68]), research suggests that EI is malleable. Scholars have documented the ability to improve EI through training and behavioral interventions [69]. Thus, while investigating attachment styles as predictors of differing communication strategies—and particularly emoji use—would provide valuable insight into the underlying individual differences that may drive such communication differences, some adults with insecure (i.e., more anxious or avoidant) attachment styles may have developed a greater degree of EI through their own life experiences. Without accounting for such a possibility, heightened EI may obscure associations between attachment styles and emoji use and produce misleading findings.

## 2. The current study

The main objective of the current study is to examine associations between attachment styles and emoji use. To expand the applicability of our findings, we also explored emotional intelligence as a moderating factor in the link between said attachment styles and emoji use. We further assessed emoji use across varying relationship contexts (i.e., friends; family; dating or romantic partners; co-workers, customers, or clients) to obtain a more informed view of how and with whom people use emojis. Based on previous research, we hypothesized that both anxious and avoidant attachments would be negatively related to emoji use frequency, while emotional intelligence would be positively related to such frequency (**Hypothesis 1).** Additionally, as prior work has shown gender differences in emoji use [5], we further hypothesized women would use emojis more frequently than men, across all social contexts (**Hypothesis 2**).

## 3. Method

The Institutional Review Board of Indiana University accepted this study as Exempt (protocol number: 1511934495; Initial approval: May 5th, 2016; Amendment approval: December 15th, 2016). Data were collected in two (2) days, between December 23rd, 2016, and December 25th, 2016. Participants electronically provided informed consent by reading a Study Information Sheet and clicking a box to agree to participate. Participation was voluntary, and participants were allowed to skip any question that they were not comfortable answering.

### 3.1. Participants and procedures

A sample of 320 adult participants ($M$ = 34.61 years, $SD$ = 11.57, range: 18–69) was recruited via Mechanical Turk. Participants completed an online survey on Qualtrics, which included questions about their demographic information, attachment styles, emotional intelligence, and emoji use (see Material section for details). To be eligible, participants had to be at least 18 years of age, fluent in English, living in the U.S., and regularly sending and receiving text-based messages in which emojis could be used. See Table 1 for demographic information.

**Table 1. Demographic information of the sample.**

| Variables | Frequency | Valid % |
|---|---|---|
| Gender | | |
| Women | 191 | 59.7 |
| Men | 123 | 38.4 |
| Transgender men | 3 | 0.9 |
| Transgender women | 1 | 0.3 |
| Other | 2 | 0.6 |
| Ethnicity | | |
| White or Caucasian | 230 | 71.9 |
| Black or African American | 38 | 11.9 |
| Asian | 30 | 9.4 |
| Hispanic, Latino or Spanish | 21 | 6.6 |
| Biracial or multiethnic | 9 | 2.8 |
| Native American or American Indian | 7 | 2.2 |
| Pacific Islander | 1 | 0.3 |
| Other | 1 | 0.3 |
| Education (highest level completed) | | |
| Bachelor's degree | 121 | 37.8 |
| Some college, but no degree | 77 | 24.1 |
| Graduate or professional degree | 45 | 14.1 |
| Associate degree | 37 | 11.6 |
| High school diploma | 27 | 8.4 |
| Vocational or technical degree | 10 | 3.1 |
| Less than high school diploma | 3 | 0.9 |
| Sexual orientation | | |
| Exclusively heterosexual/straight | 256 | 80.0 |
| Mostly heterosexual/straight | 26 | 8.1 |
| Bisexual | 17 | 5.3 |
| Exclusively homosexual/gay/lesbian | 16 | 5.0 |
| Asexual/non-sexual | 3 | 0.9 |
| Mostly homosexual/gay/lesbian | 2 | 0.6 |
| Relationship status | | |
| Married | 115 | 35.9 |
| Single and not dating anyone | 87 | 27.2 |
| In a committed romantic relationship | 78 | 24.4 |
| Casually dating one or more people | 25 | 7.8 |
| Engaged | 7 | 2.2 |
| Other | 8 | 2.5 |

### 3.2. Materials

**Demographics questionnaire.** Participants self-reported age, gender, race/ethnicity, education level, sexual orientation, and relationship status.

**Attachment style.** Participants completed the 12-item Experiences in Close Relationships–Short Form [70], responding on a 7-point Likert scale (1 = strongly disagree, 7 = strongly agree; anxious attachment $\alpha$ = .81, avoidant attachment $\alpha$ = .85).

**Emotional intelligence.** Participants completed the 30-item Trait Emotional Intelligence Questionnaire—Short Form [71], responding on a 7-point Likert scale (1 = completely disagree, 7 = completely agree; $\alpha$ = .93).

**Emoji use questionnaire.** We developed an 8-item measure of emoji use frequency across relationship types (i.e., friends; family; dating or romantic partners; and co-workers, customers, or clients). Participants responded to, "how often do you send emojis to your [relationship type]" and to, "how often do you receive emojis from your [relationship type]." Participants answered each question for all four relationship types. Responses were made on 4-point Likert scales (1 = never, 4 = very often). Emojis were defined for participants as "small digital images or icons that can be inserted into text messages, iMessages, emails, or Facebook messages," which was accompanied by four example images.

## 4. Data integrity and analytic strategy

Correlational analyses were conducted to examine the associations between attachment styles and emoji use (Hypothesis 1). Moderation analyses were also conducted to explore the effect of emotional intelligence on the association between attachment styles and emoji use. Lastly, repeated measures ANOVA with post-hoc comparisons (with Bonferroni corrections) and independent samples $t$-tests were run to test whether women would use emojis more frequently than men (Hypothesis 2). Note that, due to statistical power, only self-identified women and men were included in the gender analyses. Additionally, participants had the option to skip any question they preferred not to answer, which caused slight fluctuations in the degrees of freedom across our tests due to sporadic missing data. We assumed this data was missing at random and left it as-is, without imputing any values, as the amount of missing data was too small to reliably explore potential correlates. All analyses were run on SPSS v.27 (see Table 2 for descriptive statistics).

## 5. Results

### 5.1. Examining associations between attachment styles, emotional intelligence, and emoji use (Hypothesis 1; partly supported)

Avoidant attachment was weakly negatively related to emoji use with friends and with dating or romantic partners, but was unrelated to emoji use with other relationship partners. Anxious attachment was unrelated to emoji use across all relationship types. Notably, emotional intelligence was weakly positively related to emojis use with friends, but unrelated to emoji use with other types of relationship partners (see Table 3 for correlations). In sum, our first hypothesis —that both anxious and avoidant attachments would be negatively related to emoji use frequency, while emotional intelligence would be positively related to such emoji use frequency —was partially supported.

For a more nuanced perspective, we conducted correlations within the subset of women and men separately to investigate gender-specific patterns. Women's avoidant attachment was weakly negatively related to emoji use with friends, and negatively related to receiving emojis from family and dating or romantic partners. Women's avoidant attachment was moderately

**Table 2. Descriptive statistics of the main variables.**

| Variables | | Total | | | Women | | | Men | | |
|---|---|---|---|---|---|---|---|---|---|---|
| | | Mean | SD | SEM | Mean | SD | SEM | Mean | SD | SEM |
| Attachment styles | | | | | | | | | | |
| | Anxious | 3.39 | 1.29 | 0.07 | 3.35 | 1.25 | 0.09 | 3.45 | 1.37 | 0.12 |
| | Avoidant | 2.64 | 1.19 | 0.07 | 2.50 | 1.17 | 0.08 | 2.88 | 1.20 | 0.11 |
| Emotional intelligence | | 146.47 | 27.76 | 1.56 | 146.76 | 27.08 | 1.97 | 145.45 | 28.61 | 2.58 |
| How often do you send emojis to [. . .]? | | | | | | | | | | |
| | Friends | 3.35 | 0.78 | 0.04 | 3.47 | 0.70 | 0.05 | 3.15 | 0.87 | 0.08 |
| | Family | 3.06 | 0.90 | 0.05 | 3.22 | 0.83 | 0.06 | 2.80 | 0.96 | 0.09 |
| | Dating or romantic partners | 3.20 | 0.98 | 0.05 | 3.21 | 1.00 | 0.07 | 3.17 | 0.96 | 0.09 |
| | Co-workers, customers, or clients | 2.03 | 1.02 | 0.06 | 2.07 | 1.04 | 0.08 | 1.97 | 0.98 | 0.09 |
| How often do you receive emojis from [. . .]? | | | | | | | | | | |
| | Friends | 3.32 | 0.73 | 0.04 | 3.42 | 0.69 | 0.05 | 3.15 | 0.76 | 0.07 |
| | Family | 2.98 | 0.87 | 0.05 | 3.06 | 0.87 | 0.06 | 2.86 | 0.87 | 0.08 |
| | Dating or romantic partners | 3.06 | 0.99 | 0.06 | 2.95 | 1.04 | 0.08 | 3.20 | 0.90 | 0.08 |
| | Co-workers, customers, or clients | 2.03 | 0.96 | 0.05 | 2.01 | 0.97 | 0.07 | 2.06 | 0.94 | 0.09 |

negatively associated with sending emojis to dating or romantic partners. Men's avoidant attachment was weakly negatively related to sending emojis to dating or romantic partners; no other attachment associations emerged for men. Men's and women's anxious attachment was unrelated to emoji use. Last, emotional intelligence was weakly positively associated with women's emoji use with friends, and with men's sending emojis to dating or romantic partners.

After correction for multiple testing (i.e., Bonferroni correction), emotional intelligence did not moderate any of the relationships between attachment styles (anxious or avoidant) and the frequency of sending emojis to or receiving them from friends, family, dating or romantic partners, and co-workers, customers, or clients.

**Table 3. Correlations between main variables and emoji use frequency.**

| Variables | | | Total | | | Women | | | Men | | |
|---|---|---|---|---|---|---|---|---|---|---|---|
| | | | Attachment styles | | Emotional intelligence | Attachment styles | | Emotional intelligence | Attachment styles | | Emotional intelligence |
| | | | Anxious | Avoidant | | Anxious | Avoidant | | Anxious | Avoidant | |
| How often do you [send/receive] emojis to/from [. . .]? | Friends | Send | .02 | -.19** | .15** | -.03 | -.23** | .21** | .10 | -.08 | .06 |
| | | Receive | -.04 | -.13* | .17** | -.05 | -.22** | .23** | -.02 | .04 | .05 |
| | Family | Send | -.04 | -.11 | .08 | -.08 | -.08 | .08 | .03 | -.08 | .10 |
| | | Receive | -.07 | -.09 | .05 | -.11 | -.16* | .07 | -.00 | .04 | .05 |
| | Dating or romantic partner | Send | .06 | -.29** | .07 | .07 | -.30** | -.02 | .05 | -.26** | .19* |
| | | Receive | .07 | -.15** | .04 | .14 | -.19* | -.04 | -.02 | -.15 | .15 |
| | Co-workers, customers, or clients | Send | .01 | .00 | .01 | -.05 | .02 | -.01 | .10 | -.03 | -.04 |
| | | Receive | .02 | .09 | .02 | -.02 | .06 | .04 | .09 | .11 | -.14 |

Note.
*$p < .05$,
**$p < .01$.

## 5.2. Examining gender differences in emoji use frequency across relationship types (Hypothesis 2; partly supported)

Repeated measures ANOVA with Bonferroni corrections revealed a large statistically significant main effect of relationship types on the frequency of sending, $F(2.77, 863.12) = 190.54$, $p < .001$, $\eta2 = .38$, and receiving, $F(2.90, 904.16) = 160.49$, $p < .001$, $\eta2 = .34$, emojis. In both cases, Mauchly's Test of Sphericity indicated that the assumption of sphericity had been violated (sending: $\chi2(5) = 44.36$, $p < .001$; receiving: $\chi2(5) = 18.06$, $p < .001$), so Greenhouse-Geisser corrections were used. When it comes to sending emojis, all Bonferroni corrected pairwise comparisons were significant ($p < .001$), except for the difference between the frequency of emojis sent to friends and dating or romantic partners—the frequency of emojis sent to those types of relationships did not differ. When it comes to receiving emojis, all Bonferroni corrected pairwise comparisons were significant ($p < .001$), except for the difference between the frequency of emojis received from family and dating or romantic partners—again, the frequency between those types of relationships did not differ. Independent samples $t$-tests then revealed that, compared to men, women send more emojis to their friends, $t(312) = 3.65$, $p < .001$, $d = .41$, and family, $t(312) = 4.05$, $p < .001$, $d = .47$, as well as receive more emojis from friends, $t(312) = 3.33$, $p < .001$, $d = .37$. There were no significant gender differences in sending emojis to dating/romantic partners or co-workers/customers/clients, or in receiving emojis from anyone except friends. See Fig 1 for details and Table 2 for descriptive statistics.

## 6. Discussion

These results show several correlations between attachment style, emotional intelligence, and emoji use. These associations differ by gender and relationship contexts (i.e., friends; family; dating/romantic partners; or customers/clients/co-workers). For women, higher levels of attachment avoidance were correlated with sending and receiving emojis less often with friends and dating or romantic partners. For men, higher levels of attachment avoidance was correlated with sending fewer emojis to such partners but not in receiving them less often. Emotional intelligence was also only weakly positively correlated to women's emoji use with friends. Last, women used emojis more than men, but this difference seems to be specific to interactions with friends and family.

### 6.1. Interpretations

Although the strength of these associations is relatively modest, these results are consistent with previous research on emoji use and previous gender differences found between women's

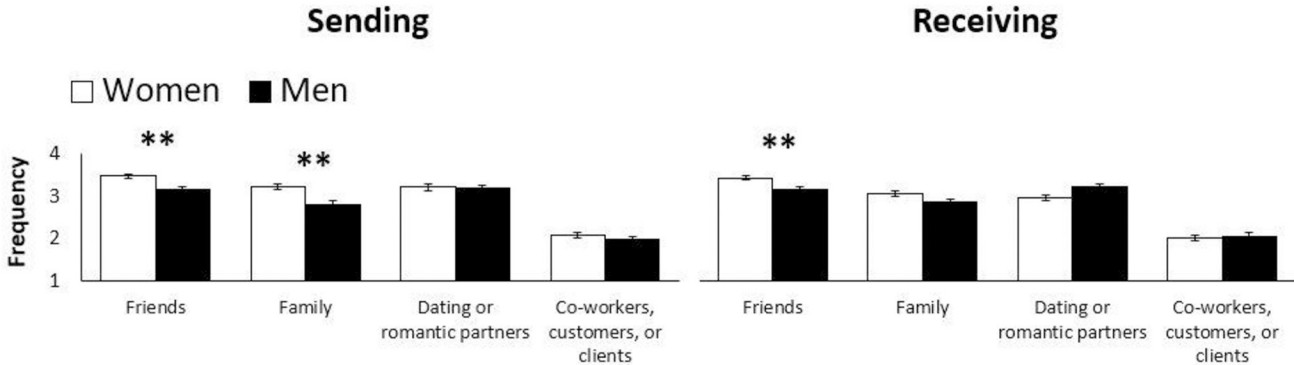

**Fig 1. Mean frequency of emoji use across genders and relationship types.** Error bars represent the standard errors of the means. *Note.* **$p < .01$.

and men's usage of emojis (cf., [5]). Notably, these results expand research on emoji use as related to individual differences pertaining to communication skills and interpersonal relationships. Our results could suggest that those who are likely to experience more discomfort with closeness and intimacy may also distance themselves from emotional conveyance within the context of CMC [72,73]. Our results could further suggest that people's ability to process emotions may influence virtual communication with different targets, such as friends or intimate partners.

In addition, the gender differences found in this study suggest that women may be more emotionally expressive, want to convey more complex meaning, and/or aim to be more precise in their CMC, particularly with friends and family [1,5,74]. However, although we did not specifically ask participants to consider expectations of emoji use—particularly along gender lines —we acknowledge that women may also feel societal pressure to use emojis in accordance with gendered expectations for communication style, as noted in previous research on these differences (cf., [75–79]). Specifically, Tannen [79] proposes that women are expected to use more "rapport" talk, which focuses on relationship building as opposed to men who are expected to use more "report" talk, or focus on task-oriented communication. Together, these findings suggest that our individual differences may translate into CMC and how people leverage the tools at their disposal in distanced communication. In turn, this may influence the relationship dynamics that people have with, for instance, friends, family, intimate partners, and work connections.

## 6.2. Limitations and strengths

This study has methodological and sampling limitations. Methodologically, this study is limited by its cross-sectional nature, self-report measures, the current lack of psychometrically validated measures of emoji use, and thus a reliance on assessments of emoji use frequency (vs. the specific emojis used or perceptions of emojis). This study also uses 4-point Likert scales rather than more expansive 7-point scales which are known to provide more accuracy in data collection (cf., [80]).

In terms of sampling, this study employed an online convenience sample mostly composed of white, educated, married, and English-speaking heterosexual individuals with a limited number of minority-identifying individuals included. Based on the data collection window for this study—December 23rd to 26th—the possibility of selection bias should further be noted. Indeed, amongst the U.S.-based population who participated in this study, these dates coincide with a period of religious and/or family-oriented winter holiday celebrations in which individuals may engage in particular interpersonal interactions that influence both their CMC behavior and expectations.

Nonetheless, this is the first study to examine emoji use in tandem with key individual characteristics related to people's communication abilities and interpersonal relationships (i.e., attachment styles and emotional intelligence) across various interpersonal contexts. As such, it contributes to the growing literature on CMC, emoji use, and individual characteristics. It also provides novel insights into *who* may use emojis more frequently, as well as *with whom* and in what relational context. Finally, our work opens new research avenues at the intersections of psychology, CMC, and the study of attachment and emotional intelligence.

## 6.3. Future directions

While this study was primarily limited to exploring emoji use across women and men, future research should examine potential differences in larger, more diverse and representative samples based on other dimensions (e.g., gender/sex, sexual orientation, or relationship status).

Future research should also extend the data collection window, acknowledge its implication of the recruited samples, or consider examining broader, more longitudinal use of emojis. Moreover, this line of research should examine the type of emojis being exchanged, the content of the message that accompanies them, and the perception of these exchanges across genders and relationship types. Notably, this work would benefit from accessing digital records of emoji exchanges to avoid relying on self-report of past interactions. This research is important given the prevalence of distanced communications in our lives.

## Author Contributions

**Conceptualization:** Simon Dubé, Amanda N. Gesselman, Vivian P. Ta-Johnson, Justin R. Garcia.

**Data curation:** Amanda N. Gesselman.

**Formal analysis:** Simon Dubé.

**Writing – original draft:** Simon Dubé, Amanda N. Gesselman, Ellen M. Kaufman, Margaret Bennett-Brown.

**Writing – review & editing:** Simon Dubé, Vivian P. Ta-Johnson, Justin R. Garcia.

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
