## [Decision Letter · Decision Letter 0]

5 Dec 2023

PONE-D-23-28745Beyond words: Relationships between emoji use, personality traits, attachment style, and emotional intelligencePLOS ONE

Dear Dr. Dubé,

Thank you for submitting your manuscript to PLOS ONE. After careful consideration, we feel that it has merit but does not fully meet PLOS ONE’s publication criteria as it currently stands. Therefore, we invite you to submit a revised version of the manuscript that addresses the points raised during the review process.

Please attend to the lack of information in your manuscript, and also attend to all the reviewer's comments to improve your manuscript and send it to a second round.

We look forward to receiving your revised manuscript.

Kind regards,

Daniela Moctezuma

Academic Editor

PLOS ONE

2. Please include a copy of Tables 1, 2, 3, and 4 which you refer to in your text on pages 10, 11, 12, and 13.

Reviewers' comments:

Reviewer's Responses to Questions

**Comments to the Author**

1. Is the manuscript technically sound, and do the data support the conclusions?

Reviewer #1: Partly

Reviewer #2: No

2. Has the statistical analysis been performed appropriately and rigorously? 

Reviewer #1: Yes

Reviewer #2: I Don't Know

3. Have the authors made all data underlying the findings in their manuscript fully available?

Reviewer #1: Yes

Reviewer #2: Yes

4. Is the manuscript presented in an intelligible fashion and written in standard English?

Reviewer #1: Yes

Reviewer #2: Yes

5. Review Comments to the Author

Reviewer #1: The paper investigates how the use of emoji is correlated to personality traits, attachment styles, and emotional intelligence across genders and relationship types, surveying for measures of emoji use frequency, also including the Big Five model.

The paper finds that extraversion, agreeableness, conscientiousness, openness to experience, and emotional intelligence are positively correlated with the frequency of use of emojis, while avoidant attachment is slightly negatively correlated with it, varying by gender and relationship type (e.g., women use emojis more than men with friends and family).

The research is relevant and novel, and would benefit of the following considerations.

1: PSYCHOLOGICAL MODELS AND CONCEPTS. Despite being still studied, the Big Five model is often considered limited from a computational point of view, and also criticized from a psychological perspective: the personality traits are considered too heterogeneous and not characteristic; strongly differing with culture; and the related official survey is modeled on a narrow sample with evident psychometric biases, which may reflect in its application to any sample. Such limitations should at least be cited clearly in the paper, in a proper section, if not removing this model from the study.

On the other hand, attachment theory is an up-to-date, more solid and accepted one, thus more relevant, but weakly introduced with too generalized beliefs, such as "Specifically, warm and responsive child-caregiver relationships facilitate the development of a secure attachment style, whereas inconsistent and unresponsive child-caregiver relationships facilitate the development of insecure attachment styles (Bowlby 1969/1982). Insecure attachment styles are categorized as anxious or avoidant (Brennan et al., 1998)", which stand only a superficial and limiting description, which could lead to misunderstandings and new biases in the perception of non-expert readers. A more in-depth description of the main standing points of the theory is necessary to understand the contribution on the topic.

Also, speaking of "emotional stability" out of the context given in the original research is not correct here, because the term "emotion" conveys a temporary state of the human neurological system, and not a stable state. Thus, the concept should be provided in different terms.

2: BIASES. Some information given in the state-of-the-art analysis is evidently including gender bias, such as "in studies of email correspondence from institutional leaders, emojis are associated with positive perceptions of likability and effectiveness for men, but viewed as positive but less appropriate for women (Riordan & Glikson, 2020)", or selection bias, such as "Liu and Sun (2020) found that extraversion and neuroticism (i.e., low emotional stability) were respectively negatively and positively associated with using emojis to avoid awkwardness", which is not useful if given as a knowledge base without highlighting the limits of biases that a strong ethical fundation should avoid to prolong.

A further selection bias to be highlighted could be because "Data were collected in two (2) days, between December 23rd, 2016, and December 25th , 2016", which is a period in which usually a large part of the population lives Christmas (as a religious or family-gathering event), especially in the U.S:, where the study had been carried. Thus, people alone in front of the PC, ready to reply to the survey on Christmas eve and day, may be already related to particular personal or interpersonal situations affecting their feedback.

It is important to refrain that biases present in a study don't make the study bugged, but only limited, and such limitations should be clearly presented to the reader, for a clearer and more precise understanding of results.

3: METHODS. The choice of Likert scales should be explained, in particular for the 4-items scale, based that an agreement/disagreement Likert scale is generally built on 5 or 7 modes, to avoid assumptions on intermediate states.

In the introduction, it is stated that the only dimension on which the use of emojis is studied is gender, and in the results gender is again the only demographic data on which results are discussed. Results could be discussed by different subgrouping, whenever even a partial set of classes is sufficiently represented in the data, as it happens already in gender with woman and man being represented, while gender-neutral people are unbalanced.

3: CLARITY OF DIFFERENCE BETWEEN CORRELATION AND CAUSATION. Among the main issues in this paper is apparently in the narrative of correlation being "relation", "associations", then -correctly- "correlations", but at the end becoming generalized statements. For instance, a weakly generalized statement is "These results show several associations between emoji use, personality traits, attachment style, and emotional intelligence, albeit with small effects", where the focus of the review is on "effects" which semantics leads more to a cause-effect relations.

Another more generalized statement is "our results further suggest that people’s ability to process emotions may influence virtual communication", where emotions and people's ability to process them largely varies in time, situation, personal and environmental independent variables that are not considered or cannot be caught in this kind of analysis. In general, the meaning of correlation should be more clear, avoiding drawing inferences that overstate its value.

4. REFERENCE LIST. The reference list may be a bit more up-to-date, since the given studies refer to a broad class of elements (emoticons, emojis, stickers) having particular differences, with a narrow application (e.g., positive/negative sentiment), while more recent studies focus more on affective computing and emotion recognition.

Please add in the submission platform the information about the Format for specific study types, Human Subject Research (involving human

participants): Indicate the form of consent obtained (written/oral) or the reason that consent was not obtained (e.g. the data were analyzed anonymously)

Reviewer #2: The paper proposes an analysis of the correlation between some personality traits, attachment style, emotional intelligence, and the usage of emojis.

I think the paper is very interesting, nevertheless, the manuscript is incomplete because all the table's contents are missing and without this information, it is difficult to evaluate your work. Also, I think it will be more interesting to exploit the data a little more because your analysis is a little simple. For instance, analyzing the correlation leaving out one trait (or group of), using features selection techniques to analyze the relevance of some traits related to classify (for instance) the type of emoji used or what EI they have, just to mention some ideas.

6. PLOS authors have the option to publish the peer review history of their article (what does this mean?). If published, this will include your full peer review and any attached files.

Reviewer #1: No

Reviewer #2: No

---

## [Author Response · Author response to Decision Letter 0]

5 Feb 2024

Response to the Reviewers

Emily Chenette, PhD

Editor-in-Chief, PLOS ONE

Janvier 27th, 2025

Dear Dr. Chenette

 We are pleased to resubmit our manuscript re-titled, “Beyond words: Relationships between emoji use, attachment style, and emotional intelligence” for consideration to PLoS ONE as an empirical article. We would like to thank the editor and the reviewers for their constructive feedback. We believe that our manuscript is much better now that we have incorporated their suggestions. We are also pleased to hear that the editor and reviewers feel that our paper is novel, relevant, and interesting. In what follows, please find a list of responses to each comment in the order that the comments were received.

Comments from Reviewer 1

Comment 1

PSYCHOLOGICAL MODELS AND CONCEPTS. Despite being still studied, the Big Five model is often considered limited from a computational point of view, and also criticized from a psychological perspective: the personality traits are considered too heterogeneous and not characteristic; strongly differing with culture; and the related official survey is modeled on a narrow sample with evident psychometric biases, which may reflect in its application to any sample. Such limitations should at least be cited clearly in the paper, in a proper section, if not removing this model from the study.

Response 1

Thank you for pointing out this weakness in our paper. Based on this comment, we have made the decision to cut the Big Five from our paper entirely. Our paper now primarily focuses on emoji use, attachment styles, and emotional intelligence, with the latter also explored as a potential moderator. We believe that this change streamlines our article, making it more cohesive and providing a clearer, more focused contribution to the area of study as a whole. 

Comment 2

On the other hand, attachment theory is an up-to-date, more solid and accepted one, thus more relevant, but weakly introduced with too generalized beliefs, such as "Specifically, warm and responsive child-caregiver relationships facilitate the development of a secure attachment style, whereas inconsistent and unresponsive child-caregiver relationships facilitate the development of insecure attachment styles (Bowlby 1969/1982). Insecure attachment styles are categorized as anxious or avoidant (Brennan et al., 1998)", which stand only a superficial and limiting description, which could lead to misunderstandings and new biases in the perception of non-expert readers. A more in-depth description of the main standing points of the theory is necessary to understand the contribution on the topic.

Response 2

This is an excellent point and we appreciate the opportunity to expand upon attachment theory. We have eliminated the too-generalized sentences and added a more robust section to delve into the main points of the theory and the current research associated with attachment styles and the theory as a whole. 

Comment 3

Also, speaking of "emotional stability" out of the context given in the original research is not correct here, because the term "emotion" conveys a temporary state of the human neurological system, and not a stable state. Thus, the concept should be provided in different terms.

Response 3

Thank you. That was indeed a pertinent comment. However, since we removed personality traits from this article, we believe this comment no longer applies to the new manuscript. 

Comment 4

BIASES. Some information given in the state-of-the-art analysis is evidently including gender bias, such as "in studies of email correspondence from institutional leaders, emojis are associated with positive perceptions of likability and effectiveness for men, but viewed as positive but less appropriate for women (Riordan & Glikson, 2020)", or selection bias, such as "Liu and Sun (2020) found that extraversion and neuroticism (i.e., low emotional stability) were respectively negatively and positively associated with using emojis to avoid awkwardness", which is not useful if given as a knowledge base without highlighting the limits of biases that a strong ethical fundation should avoid to prolong.

A further selection bias to be highlighted could be because "Data were collected in two (2) days, between December 23rd, 2016, and December 25th , 2016", which is a period in which usually a large part of the population lives Christmas (as a religious or family-gathering event), especially in the U.S:, where the study had been carried. Thus, people alone in front of the PC, ready to reply to the survey on Christmas eve and day, may be already related to particular personal or interpersonal situations affecting their feedback.

It is important to refrain that biases present in a study don't make the study bugged, but only limited, and such limitations should be clearly presented to the reader, for a clearer and more precise understanding of results.

Response 4

Thank you for your note about biases within the existing literature. We have added the following line to the introduction section to emphasize the presence of these biases: “These studies demonstrate that there may not only be contextual and gender-based differences in frequency and type of use of emojis, but also potential biases towards emoji use (e.g., based on the user’s gender).” 

We have also added the following line to the Interpretation section to underline how this consideration relates to our findings on gender differences in emoji use: 

“However, although we did not specifically ask participants to consider expectations of emoji use—particularly along gender lines—we acknowledge that women may also feel societal pressure to use emojis in accordance with gendered expectations for communication style, as noted in previous research on these differences (c.f.,, Burgoon & Dillman, 1995; Dovidio et al., 1988; Holmes, 1995; Rogers, 1989; Tannen, 1990). Specifically, Tannen (1990) proposes that women are expected to use more “rapport” talk, which focuses on relationship building as opposed to men who are expected to use more “report” talk, or focus on task-oriented communication.”

With regard to selection bias on our part, we acknowledge this possibility in our Limitations section:

“Based on the data collection window for this study—December 23rd to 26th—the possibility of selection bias should further be noted. Indeed, amongst the U.S.-based population who participated in this study, these dates coincide with a period of religious and/or family-oriented winter holiday celebrations in which individuals may engage in particular interpersonal interactions that influence both their CMC behavior and expectations.”

We further added the following content to the Future Direction section:

“While this study was primarily limited to exploring emoji use across women and men, future research should examine potential differences in larger, more diverse and representative samples based on other dimensions (e.g., gender/sex, sexual orientation, or relationship status). Future research should also extend the data collection window, acknowledge its implication of the recruited samples, or consider examining broader, more longitudinal use of emojis.”

Comment 5

METHODS. The choice of Likert scales should be explained, in particular for the 4-items scale, based that an agreement/disagreement Likert scale is generally built on 5 or 7 modes, to avoid assumptions on intermediate states.

Response 5

When designing the survey, we elected to present 4-point scales as a method of forcing participants to ‘choose a side’ rather than choosing a neutral scale point. However, we recognize that this limits our generalizability, especially as some research suggests that more expansive scales, and particularly 7-point scales, are most accurate for assessing participants’ opinions and behaviors. We have added the presence of 4-point scales as a limitation: “This study also uses 4-point Likert scales rather than more expansive 7-point scales which are known to provide more accuracy in data collection (cf., Finstad, 2010).”

Comment 6

In the introduction, it is stated that the only dimension on which the use of emojis is studied is gender, and in the results gender is again the only demographic data on which results are discussed. Results could be discussed by different subgrouping, whenever even a partial set of classes is sufficiently represented in the data, as it happens already in gender with woman and man being represented, while gender-neutral people are unbalanced.

Response 6

We absolutely agree that this topic presents many opportunities for future study along other dimensions that were not highlighted within our study. However, as noted in our Limitations section, the imbalance in participation along these parameters prevented rigorous analysis within this particular research. We have indicated the following in the Limitations section: “In terms of sampling, this study employed an online convenience sample mostly composed of white, educated, married, and English-speaking heterosexual individuals with a limited number of minority-identifying individuals included.” We also added the following line to the Future Directions section: “While this study was primarily limited to exploring emoji use across women and men, future research should examine potential differences in larger, more diverse and representative samples based on other dimensions (e.g., gender/sex, sexual orientation, or relationship status).”

Comment 7

CLARITY OF DIFFERENCE BETWEEN CORRELATION AND CAUSATION. Among the main issues in this paper is apparently in the narrative of correlation being "relation", "associations", then -correctly- "correlations", but at the end becoming generalized statements. For instance, a weakly generalized statement is "These results show several associations between emoji use, personality traits, attachment style, and emotional intelligence, albeit with small effects", where the focus of the review is on "effects" which semantics leads more to a cause-effect relations.

Another more generalized statement is "our results further suggest that people’s ability to process emotions may influence virtual communication", where emotions and people's ability to process them largely varies in time, situation, personal and environmental independent variables that are not considered or cannot be caught in this kind of analysis. In general, the meaning of correlation should be more clear, avoiding drawing inferences that overstate its value.

Response 7 

Thank you very much for this comment. Across the manuscript—and especially in the discussion—we have adjusted our wording to indicate/emphasize correlation not causation. We have inserted additional qualifiers to be more mindful to not overstate the value of the study. 

Comment 8

REFERENCE LIST. The reference list may be a bit more up-to-date, since the given studies refer to a broad class of elements (emoticons, emojis, stickers) having particular differences, with a narrow application (e.g., positive/negative sentiment), while more recent studies focus more on affective computing and emotion recognition.

Response 8

This is absolutely true. We appreciate the attention to detail and have included more recent studies across domains that become the main focus of this manuscript (i.e., attachment theory and emotional intelligence). To elaborate on and render justice to attachment theory and gendered communication patterns, we still elected to keep and add several, more foundational studies for a well-rounded background review.

Comments from Reviewer 2

Comment 1

I think it will be more interesting to exploit the data a little more because your analysis is a little simple. For instance, analyzing the correlation leaving out one trait (or group of), using features selection techniques to analyze the relevance of some traits related to classify (for instance) the type of emoji used or what EI they have, just to mention some ideas.

Response 1

Thank you very much for this relevant comment. Based on it, we decided to explore whether emotional intelligence moderated the link between attachment style and emoji use. While the results were non-significant, the value of null results should not be underestimated.

Comments from the Editor

Comment 1

Please add in the submission platform the information about the Format for specific study types, Human Subject Research (involving human participants): Indicate the form of consent obtained (written/oral) or the reason that consent was not obtained (e.g. the data were analyzed anonymously)

Response 1

Under Method, please note the following line specifying how consent was obtained: “Participants electronically provided informed consent by reading a Study Information Sheet and clicking a box to agree to participate.” 

We hope these modifications are to your liking. If there is anything else, please do not hesitate to contact us. Again, we would like to thank the editor and the reviewers for their efficiency and feedback. Your time and help in presenting our paper to the world is extremely appreciated.

Sincerely,

The Corresponding Author

---

## [Decision Letter · Decision Letter 1]

22 Mar 2024

PONE-D-23-28745R1Beyond words: Relationships between emoji use, attachment style, and emotional intelligencePLOS ONE

Dear Dr. Dubé,

Thank you for submitting your manuscript to PLOS ONE. After careful consideration, we feel that it has merit but does not fully meet PLOS ONE’s publication criteria as it currently stands. Therefore, we invite you to submit a revised version of the manuscript that addresses the points raised during the review process.

We look forward to receiving your revised manuscript.

Kind regards,

Daniela Moctezuma

Academic Editor

PLOS ONE

Journal Requirements:

Reviewers' comments:

Reviewer's Responses to Questions

**Comments to the Author**

1. If the authors have adequately addressed your comments raised in a previous round of review and you feel that this manuscript is now acceptable for publication, you may indicate that here to bypass the “Comments to the Author” section, enter your conflict of interest statement in the “Confidential to Editor” section, and submit your "Accept" recommendation.

Reviewer #1: (No Response)

Reviewer #2: All comments have been addressed

2. Is the manuscript technically sound, and do the data support the conclusions?

Reviewer #1: Yes

Reviewer #2: (No Response)

3. Has the statistical analysis been performed appropriately and rigorously? 

Reviewer #1: N/A

Reviewer #2: (No Response)

4. Have the authors made all data underlying the findings in their manuscript fully available?

Reviewer #1: Yes

Reviewer #2: No

5. Is the manuscript presented in an intelligible fashion and written in standard English?

Reviewer #1: Yes

Reviewer #2: Yes

6. Review Comments to the Author

Reviewer #1: The previous comments have been mostly addressed, except for the reference list: with the new deletions and additions, is now very focused on couple attachment (where attachment is related to a couple in a loving relationship), whereas in the state of the art the term is more related to parental attachment, which is relevant to the following relationships in life, including loving relationship and friendship, as well as any attachment with a leading role.

This seems a little off-topic. The various papers on loving relationships are not related to "affective computing" as the reviewer asked earlier. Affective computing refers to techniques that use machine learning, data analysis and artificial intelligence for emotion recognition and affective states. A state-of-the-art review, at least an introductory one, should be included.

Reviewer #2: No more comments, I think the authors made the asked changes and the manuscript is in better shape now.

7. PLOS authors have the option to publish the peer review history of their article (what does this mean?). If published, this will include your full peer review and any attached files.

Reviewer #1: No

Reviewer #2: No

---

## [Author Response · Author response to Decision Letter 1]

4 May 2024

Response to the Reviewers

Emily Chenette, PhD

Editor-in-Chief, PLOS ONE

May 6th, 2025

Dear Dr. Chenette

 We are pleased to resubmit our manuscript re-titled, “Beyond words: Relationships between emoji use, attachment style, and emotional intelligence” for consideration to PLoS ONE as an empirical article. We would like to thank the editor and the reviewers for their constructive feedback. We believe that our manuscript is much better now that we have incorporated their suggestions. We are also pleased to hear that the editor and reviewers feel that our paper is novel, relevant, and interesting. In what follows, please find a list of responses to each comment in the order that the comments were received.

Comments from Reviewer 1

Comment 1: The previous comments have been mostly addressed, except for the reference list: with the new deletions and additions, is now very focused on couple attachment (where attachment is related to a couple in a loving relationship), whereas in the state of the art the term is more related to parental attachment, which is relevant to the following relationships in life, including loving relationship and friendship, as well as any attachment with a leading role.

Response 1: Thank you. We have now expanded the introduction to more broadly cover the links between attachment and relationships with family members, co-workers, and friends.

Comment 2: This seems a little off-topic. The various papers on loving relationships are not related to "affective computing" as the reviewer asked earlier. Affective computing refers to techniques that use machine learning, data analysis and artificial intelligence for emotion recognition and affective states. A state-of-the-art review, at least an introductory one, should be included.

Response 2: Thank you for this comment. We re-integrated in the introduction a more thorough overview about the emoji use literature, which now includes an introductory paragraph about its links with affective computing. Given that this is not the primary topic of this article, we believe that this paragraph should be sufficient and that should readers want to learn more about such a specific

We hope these modifications allows us to present this manuscript to a wider audience.

Sincerely,

The Corresponding Author

---

## [Decision Letter · Decision Letter 2]

2 Jul 2024

Beyond words: Relationships between emoji use, attachment style, and emotional intelligence

PONE-D-23-28745R2

Dear Dr. Dubé,

We’re pleased to inform you that your manuscript has been judged scientifically suitable for publication and will be formally accepted for publication once it meets all outstanding technical requirements.

Kind regards,

Daniela Moctezuma

Academic Editor

PLOS ONE

Additional Editor Comments (optional):

The reviewers agreed with your modifications and the answers given by their comments and recommendations.

Reviewers' comments:

Reviewer's Responses to Questions

**Comments to the Author**

1. If the authors have adequately addressed your comments raised in a previous round of review and you feel that this manuscript is now acceptable for publication, you may indicate that here to bypass the “Comments to the Author” section, enter your conflict of interest statement in the “Confidential to Editor” section, and submit your "Accept" recommendation.

Reviewer #2: All comments have been addressed

Reviewer #3: All comments have been addressed

2. Is the manuscript technically sound, and do the data support the conclusions?

Reviewer #2: Yes

Reviewer #3: Yes

3. Has the statistical analysis been performed appropriately and rigorously? 

Reviewer #2: Yes

Reviewer #3: Yes

4. Have the authors made all data underlying the findings in their manuscript fully available?

Reviewer #2: No

Reviewer #3: Yes

5. Is the manuscript presented in an intelligible fashion and written in standard English?

Reviewer #2: Yes

Reviewer #3: Yes

6. Review Comments to the Author

Reviewer #2: Everything is attended to as was suggested. The current version of the manuscript is ready to be published, in my opinion.

Reviewer #3: The last comments have been addressed; the literature state-of-the-art has been updated. The new text in the introduction allows us to contextualize the importance of the emoji for the impact of communication and reduce uncertainty.

7. PLOS authors have the option to publish the peer review history of their article (what does this mean?). If published, this will include your full peer review and any attached files.

Reviewer #2: No

Reviewer #3: No

---

## [Editor Report · Acceptance letter]

4 Nov 2024

PONE-D-23-28745R2 

PLOS ONE

Dear Dr. Dubé, 

I'm pleased to inform you that your manuscript has been deemed suitable for publication in PLOS ONE. Congratulations! Your manuscript is now being handed over to our production team.

Kind regards, 

on behalf of

Dr. Daniela Moctezuma 

Academic Editor

PLOS ONE